# Presenilin Deficiency Results in Cellular Cholesterol Accumulation by Impairment of Protein Glycosylation and NPC1 Function

**DOI:** 10.3390/ijms25105417

**Published:** 2024-05-16

**Authors:** Marietta Fabiano, Naoto Oikawa, Anja Kerksiek, Jun-ichi Furukawa, Hirokazu Yagi, Koichi Kato, Ulrich Schweizer, Wim Annaert, Jongkyun Kang, Jie Shen, Dieter Lütjohann, Jochen Walter

**Affiliations:** 1Department of Neurology, Universitätsklinikum Bonn, 53127 Bonn, Germany; 2Institut für Biochemie und Molekularbiologie, Universitätsklinikum Bonn, Rheinische Friedrich-Wilhelms-Universität Bonn, 53115 Bonn, Germany; 3Institute of Clinical Chemistry and Clinical Pharmacology, Universitätsklinikum Bonn, 53127 Bonn, Germany; 4Department of Orthopedic Surgery, Hokkaido University Graduate School of Medicine, Sapporo 060-8638, Japan; 5Division of Glyco-Systems Biology, Institute for Glyco-Core Research, Tokai National Higher Education and Research System, Nagoya 466-8550, Japan; 6Graduate School of Pharmaceutical Sciences, Nagoya City University, Nagoya 467-8603, Japan; 7Exploratory Research Center on Life and Living Systems (ExCELLS), National Institutes of Natural Sciences, Okazaki 444-8787, Japan; 8Institute for Molecular Science, National Institutes of Natural Sciences, Okazaki 444-8585, Japan; 9Laboratory for Membrane Trafficking, VIB-Center for Brain and Disease Research, KU Leuven, 3000 Leuven, Belgium; 10Department of Neurosciences, KU Leuven, 3000 Leuven, Belgium; 11Department of Neurology, Brigham and Women’s Hospital, Harvard Medical School, Boston, MA 02115, USA; 12Program in Neuroscience, Harvard Medical School, Boston, MA 02115, USA

**Keywords:** Presenilin, cholesterol, NPC1, N-glycosylation, lysosome

## Abstract

Presenilin proteins (PS1 and PS2) represent the catalytic subunit of γ-secretase and play a critical role in the generation of the amyloid β (Aβ) peptide and the pathogenesis of Alzheimer disease (AD). However, PS proteins also exert multiple functions beyond Aβ generation. In this study, we examine the individual roles of PS1 and PS2 in cellular cholesterol metabolism. Deletion of PS1 or PS2 in mouse models led to cholesterol accumulation in cerebral neurons. Cholesterol accumulation was also observed in the lysosomes of embryonic fibroblasts from Psen1-knockout (PS1-KO) and *Psen2-KO* (PS2-KO) mice and was associated with decreased expression of the Niemann-Pick type C1 (NPC1) protein involved in intracellular cholesterol transport in late endosomal/lysosomal compartments. Mass spectrometry and complementary biochemical analyses also revealed abnormal N-glycosylation of NPC1 and several other membrane proteins in PS1-KO and PS2-KO cells. Interestingly, pharmacological inhibition of N-glycosylation resulted in intracellular cholesterol accumulation prominently in lysosomes and decreased NPC1, thereby resembling the changes in PS1-KO and PS2-KO cells. In turn, treatment of PS1-KO and PS2-KO mouse embryonic fibroblasts (MEFs) with the chaperone inducer arimoclomol partially normalized NPC1 expression and rescued lysosomal cholesterol accumulation. Additionally, the intracellular cholesterol accumulation in PS1-KO and PS2-KO MEFs was prevented by overexpression of NPC1. Collectively, these data indicate that a loss of PS function results in impaired protein N-glycosylation, which eventually causes decreased expression of NPC1 and intracellular cholesterol accumulation. This mechanism could contribute to the neurodegeneration observed in PS KO mice and potentially to the pathogenesis of AD.

## 1. Introduction

Presenilins (PS) are multi-pass membrane proteins involved in several cellular regulatory mechanisms, such as calcium homeostasis and membrane proteostasis [1,2,3]. In membrane proteostasis, PS proteins function as catalytic subunits of a membrane protease complex, γ-secretase, involved in the proteolysis of type-I membrane proteins such as Notch and amyloid precursor protein (APP) [4,5,6,7,8]. While Notch proteolysis exerts crucial functions in cell differentiation and development, APP cleavage by γ-secretase results in the secretion of amyloid β-protein (Aβ), which accumulates in the brains as senile plaques (SP), one of the pathological hallmarks of Alzheimer disease (AD) [6,9].

Interestingly, mutations in the genes encoding the two PS homologous proteins, PS1 and PS2, are a major cause of autosomal dominantly inherited forms of familial early-onset AD (EOAD) [10,11,12]. A growing body of evidence indicates that these mutations result in dysfunction of PS and/or γ-secretase (PS/γ-secretase) [13,14,15,16,17,18]. The altered function of PS/γ-secretase could also be implicated in sporadic AD [19,20], suggesting a significant involvement in AD pathogenesis. PS/γ-secretase dysfunction can contribute to AD development by inducing senile plaque formation via decreased processivity, resulting in increased production of longer aggregation-prone Aβ species such as Aβ42 [18,21,22,23] and/or by inducing neurodegeneration via Aβ-independent pathways [24,25,26,27]. However, the mechanism by which the loss of PS proteins causes neurodegeneration has not yet been fully elucidated.

One of the consequences of PS/γ-secretase dysfunction is cellular cholesterol accumulation. Genetic knock-out (KO) of *Psen1* and *Psen2* in mice and pharmacological inhibition of γ-secretase increase cellular cholesterol level by affecting several events of cellular cholesterol metabolism, i.e., lipoprotein uptake by low-density lipoprotein receptor (LDLR) and LDL receptor-related protein 1 (LRP1), lipoprotein secretion by ATP-binding cassette transporter A1 (ABCA1), and cholesterol biosynthesis by sterol regulatory element binding protein-2 (SREBP2), liver x receptor (LXR), 3-hydroxy-3-methyl-glutaryl-coenzyme A (HMG-CoA) reductase, and CYP51 [28,29,30,31,32]. Importantly, elevated cholesterol has been reported in cells expressing AD-causing PS1 or PS2 mutants, suggesting that altered cholesterol metabolism could contribute to the pathogenesis of EOAD as well [28,33,34]. Regulation of cellular cholesterol metabolism is important for cellular physiological functions such as transmembrane signaling and membrane trafficking [35,36,37]. An altered cholesterol metabolism can result in neurodegeneration, as seen in Niemann Pick disease type C (NPC), in which impairment of the cholesterol transporter Niemann-Pick type C1 (NPC1) protein induces intracellular cholesterol accumulation [38,39]. It is important to note that neurofibrillary tangle (NFT) formation, which is also a pathological hallmark of AD, is observed in the brains of NPC patients [40,41,42]. Therefore, it is important to clarify the functional relationship between PS/γ-secretase dysfunction and cellular cholesterol metabolism.

In this study, we investigate how cholesterol accumulates in genetic models of PS/γ-secretase dysfunction: cerebral cortical neurons and fibroblasts from individual PS1-KO and PS2-KO mice [43,44].

## 2. Results

### 2.1. Deletion of PS1 or PS2 Induces Cellular Cholesterol Accumulation

A previous study showed increased free cholesterol in PS1/2 conditional double KO mouse brains [28]. To examine the relative contribution of PS1 and PS2 in cholesterol accumulation, we analyzed cholesterol levels in mouse brains with conditional deletion of PS1 in excitatory forebrain neurons (PS1cKO) [43] and in mice with constitutive deletion of PS2 (PS2KO) [44]. Visualization of free cholesterol by filipin, which selectively binds to free cholesterol [45], revealed significantly increased levels of cholesterol in neurons in the cortex and hippocampal CA1 region of PS1cKO and PS2KO mice (Figure 1). This data suggests that deletion of either PS1 or PS2 induces cholesterol accumulation in cerebral neurons.

To provide further insight into the molecular mechanism underlying the cholesterol accumulation upon PS deletion, embryonic fibroblasts derived from PS1-KO and PS2-KO mice were analyzed by mass spectrometry (MS). Cholesterol and its precursors, desmosterol and lathosterol, were significantly increased in PS1- and PS2-KO mouse embryonic fibroblasts (MEFs) (Figure 2A), indicating increased biosynthesis and accumulation of cholesterol. Filipin staining combined with immunostaining for different endosomal and lysosomal compartments revealed a significant increase in the colocalization of free cholesterol with LAMP2-positive lysosomal and RAB7-positive late endosomal compartments, but not with EEA1-positive early endosomal compartments in PS1- and PS2-KO MEFs (Figure 2B).

### 2.2. Reduced Intracellular Transport and Increased Biosynthesis of Cholesterol in PS1-KO and PS2-KO MEFs

In order to understand the process that leads to the observed cholesterol accumulation in PS KO MEFs, we examined the expression level of cholesterol metabolism-related proteins by immunoblotting (Figure 3A). Lack of PS1 or PS2 similarly affected the expression level of proteins involved in intracellular cholesterol transport and cholesterol biosynthesis. The expression level of NPC2, which binds and transfers endolysosomal cholesterol to NPC1 [46,47], was increased, while NPC1 itself, which mediates the transport of cholesterol from late endosomal/lysosomal compartments to other cellular compartments, was decreased in both PS1-KO and PS2-KO MEFs (Figure 3B,C). Additionally, the expression level of the full-length and N-terminal fragments of SREBP2, a major transcriptional regulator of cellular cholesterol metabolism, was significantly increased in PS1-KO as well as in PS2-KO cells. The level of CYP51 was also significantly increased in the PS KO cell model, further supporting elevated de novo cholesterol biosynthesis [48] (Figure 3B,C). On the other hand, expression of lipoprotein receptors and ABCA1 was differentially affected in PS1 and PS2 KO cells. Expression levels of LDLR and ABCA1 were increased in PS1-KO MEFs while decreased and not different in PS2-KO MEFs (Figure 3B,C). Also, levels of LRP1 were decreased in PS2-KO MEFs (Figure 3B,C). These results suggest that enhanced cellular cholesterol biosynthesis and/or impairment of intracellular cholesterol transport from late endosomal/lysosomal compartments to other compartments can be responsible for the cellular cholesterol accumulation in PS1- or PS2-deficient cells.

mRNA expression analysis revealed increased levels of mRNA for *Ldlr* in PS2-KO, *Abca1* in PS1-KO, and *Cyp51* in both PS1-KO and PS2-KO MEFs (Figure 3D). mRNA levels of *Npc1* and *Npc2* were rather decreased in PS1-KO and PS2-KO MEFs, respectively (Figure 3D). However, the mRNA expression of other cholesterol metabolism-related genes was not significantly altered. It is worth noting that levels of some cholesterol metabolism-related proteins can be regulated by post-translational mechanisms. For example, SREBP2 can be stabilized by binding to the SREBP cleavage activating protein (SCAP), resulting in increased generation of the transcriptionally active N-terminal domain of SREBP2 by site-1 protease (S1P) and site-2 protease (S2P) [49,50]. In addition, NPC2 can be stabilized upon binding to the Nogo-B receptor or glycine N-methyltransferase [51,52]. The expression level of NPC1 can be affected by N-glycosylation, and its impairment results in an increased ER-associated protein degradation pathway [53,54]. Together, these data suggest the involvement of transcriptional and post-transcriptional/translational mechanisms in the dysregulation of protein expression in PS1- or PS2-deficient cells (Figure 3D).

### 2.3. Lack of PS1 or PS2 Affects Protein Glycosylation

Immunoblot analysis also revealed faster migration of NPC1 and LRP1 in samples from PS1-KO and PS2-KO MEFs (Figure 3A). The effect on mobility of NPC1 was stronger in PS1-KO than in PS2-KO MEFs (Figure 3A). Since these proteins undergo complex N-glycosylation, we tested the sensitivity of these proteins to PNGase F, an enzyme able to remove N-linked oligosaccharides from glycoproteins. We also analyzed the expression of other membrane glycoproteins, including N-cadherin, Lamp2, and Nicastrin. The glycosylation and maturation of Nicastrin were previously shown to be impaired in PS-deficient cells due to impaired assembly and forward transport of the γ-secretase complex [55,56,57,58,59,60]. Notably, a mobility shift was detected not only for NPC1 and LRP1, but also for other examined membrane glycoproteins in PS KO MEFs (Figure 4A). As expected, PNGase F treatment increased the migration of these proteins due to decreased molecular mass upon deglycosylation (Figure 4A). However, PS deficiency had no effect on the migration of deglycosylated proteins. These data strongly indicate that alterations in the molecular mass of these proteins from PSs KO cells are due to altered N-glycosylation. Additionally, it has been suggested that PS deficiency results in a general impairment of the N-glycosylation machinery [61].

A quantitative analysis of protein-bound glycans by mass spectrometry revealed decreased levels of the glycan Man_5_GlcNAc_2_ (M5) in major detectable glycans in both PS1-KO and PS2-KO MEFs, confirming a more general impairment of protein N-glycosylation caused by PS deficiency (Figure 4B). M5 N-glycan is generated by sequential trimming of the precursor N-glycans, i.e., Glc_2_Man_9_GlcNAc_2_, Man_9_GlcNAc_2_ (M9), Man_8_GlcNAc_2_ (M8), Man_7_GlcNAc_2_ (N7), and Man_6_GlcNAc_2_ (M6) by mannosidases, i.e., GANAB, MAN1B1, MAN1A, MAN1A2, and MAN1C1 [62]. mRNA expression analysis of genes encoding enzymes involved in the N-glycan processing showed increases of *Man1a2* and *Man1c1* in PS1-KO and PS2-KO MEFs, respectively (Figure 4C). Additionally, an increased expression of *Ganab*, an enzyme involved in the formation of M9, as well as a decreased expression of *Man1a*, responsible for the formation of M5, were detected in both PS1-KO and PS2-KO MEFs (Figure 4C). The level of *Mgat1*, which converts M5 to GlcNAcMan_5_GlcNAc_2_, was instead increased only in PS1-KO MEFs (Figure 4C). Over all, these data suggest that a decreased expression of *Man1a* could contribute to the observed reduced level of M5 in PS deficient cells.

### 2.4. Impairment of Protein Glycosylation Induces Cholesterol Accumulation

The altered protein N-glycosylation in PS-deficient cells prompted us to examine whether pharmacological impairment of glycosylation could induce cellular cholesterol accumulation. We used several alkaloids known to inhibit distinct steps in protein N-glycosylation, i.e., tunicamycin, which inhibits ALG7, a glycosyltransferase involved in the initial step of the biosynthesis of N-linked glycans; deoxynojirimycin, an inhibitor of ER α-glycosidase I and II; kifunensine, an inhibitor of ER α-1,2-mannosidase I and Golgi class I mannosidases (Golgi α-mannosidase IA, IB, and IC); and swainsonine, an inhibitor of Golgi α-mannosidase II and lysosomal α-mannosidase [63].

Treatment with tunicamycin, but not deoxynojirimycin, kifunensine, and swainsonine, induced cholesterol accumulation in late endosomal/lysosomal compartments similarly to the phenotypes observed in PS1-KO and PS2-KO MEFs (Figure 5A,B). Then, we examined whether these inhibitors affected the expression profile of the proteins by western blotting (Figure 6A,C). Treatment with the inhibitors, which did not induce intracellular cholesterol accumulation, differently affected the expression profiles of the proteins involved in cellular cholesterol metabolism. Swainsonine did not affect the expression of any examined protein; kifunensine decreased CYP51 expression and deoxynojirimycin decreased the expression of lipoprotein receptors, LDLR and LRP1, and increased ABCA1, NPC1, and NPC2 (Figure 6B,D). On the other hand, treatment with tunicamycin, which induced intracellular cholesterol accumulation, decreased NPC1 and increased NPC2 and CYP51 expression similarly to the phenotypes observed in PS1-KO and PS2-KO MEFs (Figure 6B). Tunicamycin also decreased the expression of lipoprotein receptors (LDLR and LRP1) and increased ABCA1 (Figure 6B). However, deoxynojirimycin also decreased expression of LDLR, LRP1, and increased ABCA1 and NPC2 without affecting cellular cholesterol accumulation (Figure 6B). Thus, it is unlikely that the changes in these proteins contribute to the cholesterol accumulation observed upon cell treatment with tunicamycin. In addition, deoxynojirimycin increased NPC1 expression, in contrast to what is observed with tunicamycin and in PS1-KO and PS2-KO cells. (Figure 6B).

Overall, these results suggest that an impairment of early steps of N-glycan biosynthesis contributes to the intracellular cholesterol accumulation in endolysosomal compartments in PS1 and PS2 deficient cells by altering N-glycosylation of NPC1.

### 2.5. Chaperone Induction Increases NPC1 Expression and Normalizes Endolysosomal Cholesterol Distribution in PS1-KO and PS2-KO MEFs

Protein glycosylation is involved in protein folding, and it has been shown previously that disease-associated mutations in NPC1 can cause misfolding and increase its ER-associated protein degradation [53,54]. Importantly, NPC patient cells, in which expression of functional NPC1 is decreased, show intracellular cholesterol accumulation, particularly in late endosomal/lysosomal compartments [64,65]. Thus, we next wanted to test whether promotion of protein folding could stabilize NPC1 and normalize intracellular cholesterol distribution in PS-deficient cells. PS KO MEFs were treated with arimoclomol, which has been shown to significantly reduce lysosomal accumulation of unesterified cholesterol in NPC patient fibroblasts and is currently used in clinical trials for NPC treatment [66,67]. Arimoclomol treatment indeed increased NPC1 expression levels in PS1-KO and PS2-KO MEFs (Figure 7A) and significantly attenuated intracellular cholesterol accumulation in lysosomal compartments (Figure 7B). In contrast, arimoclomol treatment of NPC1-KO cells, which show intracellular cholesterol accumulation as observed in PS-deficient cells, did not rescue the intracellular cholesterol accumulation (Figure 7C). Thus, these results suggest that the intracellular cholesterol accumulation in PS-deficient cells involves impaired N-glycosylation and destabilization of NPC1.

### 2.6. Overexpression of NPC1 Rescues Intracellular Cholesterol Accumulation in PS1-KO and PS2-KO MEFs

In order to further examine whether intracellular cholesterol accumulation in PS KO MEFs involves reduced NPC1 expression and function, we overexpressed NPC1-GFP in PS1 and PS2 KO cells. NPC1-GFP is mainly localized to LAMP2 positive compartments, indicating that the exogenously expressed NPC1-GFP is efficiently transported to lysosomes (Figure 8A). In cells expressing exogenous NPC1-GFP, filipin signals in LAMP2 compartments were significantly decreased as compared to NPC1-GFP-negative cells in both PS1KO and PS2KO MEFs (Figure 8B). These results further suggest the critical involvement of NPC1 in intracellular cholesterol accumulation upon PS1 or PS2 deficiency.

### 2.7. Effect of γ-Secretase Inhibition on NPC1 Expression

We examined whether γ-secretase dysfunction affects NPC1 expression in WT MEFs by using a γ-secretase inhibitor, *N*-[*N*-(3,5-difluorophenacetyl)-_L_-alanyl]-*S*-phenylglycine *t*-butylester (DAPT). Treatment with DAPT did not decrease NPC1 expression nor induce a mobility shift of the band in western blotting of WT MEFs (Appendix A). This result suggests that the impairment of protein glycosylation and intracellular cholesterol accumulation seen in PS KO cells are presumably independent of the γ-secretase function of PS.

## 3. Discussion

A growing body of evidence indicates that PS/γ-secretase dysfunction causes neurodegeneration. Therefore, it is crucial to elucidate the consequences of PS/γ-secretase dysfunction at the cellular level. Overall, the results of this study suggest that PS deficiency induces protein glycosylation impairment, which results in reduced levels of NPC1 and intracellular cholesterol accumulation. Since NPC1 reduction and subsequent intracellular cholesterol accumulation are causes of neurodegeneration and NFT formation in NPC, the present study provides mechanistic insight into how PS/γ-secretase dysfunction can contribute to neurodegeneration in AD development.

An involvement of PS1 in protein glycosylation has been indicated previously for several proteins such as nicastrin, APP, and TrkB [68]. Thus, our study further cumulates experimental evidence about the significant involvement of PS in general protein glycosylation. Our findings also extend previous studies, as they show that both PS homologs, PS1 and PS2, individually contribute to protein N-glycosylation. It has been proposed that PS proteins are important for glycosylation of nicastrin by direct interaction during assembly of the γ-secretase complex and its transport in the secretoy pathway [56,57,58,59,60]. It has also been proposed that PS1 directly interacts with v-ATPaseV0a1 and also facilitates transport to the Golgi compartment for further maturation by glycosylation [69], although this mechanism is under debate [70,71]. Therefore, the interaction of PS with the enzymes involved in protein glycosylation, such as asparagine-linked glycosylation (ALG) and mannosidase (MAN), could also affect protein glycosylation by modulating the activity of these enzymes. However, further investigations are required to test the direct interaction of PS proteins and the respective enzymes. As mentioned before, PS is a catalytic subunit of γ-secretase; therefore, γ-secretase dysfunction could also affect protein glycosylation. However, γ-secretase inhibition by DAPT did not induce a mobility shift or decrease of NPC1 in WT-MEFs (Appendix A). Thus, γ-secretase dysfunction may not be responsible for aberrant protein glycosylation in PS KO cells. PS proteins can also mediate intracellular membrane trafficking, cellular calcium homeostasis, and endo/lysosomal function [1,2]. Therefore, abnormalities in these mechanisms could potentially affect protein glycosylation in PS KO cells. In this regard, it is interesting to note that both PS1 and PS2 can interact with syntaxin 5 in the ER [72,73]. A mutation in syntaxin 5, which is involved in ER-Golgi and/or intra-Golgi membrane trafficking, results in defective protein glycosylation in a congenital disorder [74]. PS1 and PS2 can interact with other proteins involved in membrane trafficking [75], and perturbation of membrane trafficking affects protein glycosylation [76]. Therefore, PS1 or PS2 deficiency could result in protein glycosylation impairment by perturbing intracellular membrane trafficking.

We also found decreased mRNA expression of *Man1a* in PS KO cells that could contribute to impaired glycan biosynthesis. However, treatment with kifunensine, inhibiting MAN1B and MAN1A, did not induce endo/lysosomal cholesterol accumulation or decrease in NPC1, and thus did not mimic changes observed in PS-KO MEFs. These results suggest the contribution of other mechanisms to the endo/lysosomal cholesterol accumulation in PS-deficient cells. On the other hand, among the examined glycosylation inhibitors, tunicamycin mimicked endo/lysosomal cholesterol accumulation, NPC1 decrease, and CYP51 increase in WT MEFs, resembling the changes observed in PS-KO MEFs. These data suggest an impairment of early steps of N-glycan biosynthesis may contribute to endo/lysosomal cholesterol accumulation in PS-deficient cells. Aberrant glycosylation may cause misfolding and accelerated ER-associated degradation of NPC1, thereby lowering cholesterol transport in endo/lysosomal compartments [53,54]. Indeed, cell treatment with the chaperone inducer arimoclomol increased levels of NPC1 and attenuated lysosomal cholesterol accumulation (Figure 7A,B). However, PS deficiency can result in an increase in cellular cholesterol level by several additional pathways, including Aβ-induced stimulation of HMG-CoA reductase activity [28], increased uptake of cholesterol/lipoproteins due to increased expression of LDLR/LRP1 [29], and transcriptional upregulation of cholesterol biosynthetic enzymes [30,32]. Therefore, further elucidation of the causal relationship between protein glycosylation impairment and alteration of the expression of the proteins involved in cellular cholesterol metabolism is needed in the future.

Several studies with mouse models demonstrated that loss of function of PS/γ-secretase results in neurodegeneration independent of Aβ accumulation [25,26,27] and proposed different underlying mechanisms, including impairment of intracellular vesicle trafficking and growth factor signaling [77]. It was also proposed that increased phosphorylation and aggregation of tau mediate PS/γ-secretase-dependent neurodegeneration [25,78]. Here, we show that PS deficiency also impairs the function of NPC1 and intracellular cholesterol accumulation, which could also contribute to neurodegeneration. In that regard, it is interesting that both diseases, NPC and AD, present with neurofibrillary tangles, suggesting that tau accumulation could induce neurodegeneration in both diseases, although potentially triggered by different upstream events.

The present results further show that the loss of either PS homolog, PS1 or PS2, results in protein glycosylation impairment and intracellular cholesterol accumulation. In contrast to the situation of double KO for PS1 and PS2, deletion of PS1 or PS2 maintains the activity of the remaining γ-secretase containing PS2 or PS1, respectively, and thus Aβ production. This could be an important point since protein glycosylation impairment and intracellular cholesterol accumulation could affect cellular Aβ production. Actually, it has been reported that cellular cholesterol accumulation upon NPC1 dysfunction results in an increase in Aβ production [79,80,81]. Protein glycosylation impairment can also increase Aβ production since N-glycosylation impairment of APP can increase cellular Aβ production [82]. However, it has been reported that overall cellular protein glycosylation impairment does not affect or, rather, can result in a decrease in cellular Aβ production [83,84]. Therefore, the effect of protein glycosylation impairment on cellular Aβ production remains to be further elucidated. Alternatively, or additionally, cellular cholesterol accumulation caused by loss of PS1 or PS2 can influence Aβ aggregation at the cell membrane. Cholesterol accelerates Aβ aggregation by direct interaction [85] or by affecting the co-existing GM1-gangliosides, which may eventually promote the interaction of Aβ with GM1-ganglioside, a potent inducer of Aβ aggregation on the membrane [86,87,88]. Indeed, intracellular cholesterol accumulation by NPC1 dysfunction or reduction in vitro and in the human brain, respectively, has been shown to induce Aβ aggregation [79,89,90,91]. Thus, the intracellular cholesterol accumulation due to PS deficiency may contribute to neurodegeneration not only in an Aβ-independent manner but also in an Aβ-dependent manner.

As the present study only applied PS-KO MEFs as a cellular model to characterize the molecular mechanisms underlying the intracellular cholesterol accumulation caused by PS deficiency, it would be important to validate the present findings in different cell types of the brain, including neuronal and glial cells. Although our data revealed cholesterol accumulation in cortical and CA1 hippocampal neurons of PS1cKO and PS2KO mice, the involvement of protein N-glycosylation and NPC1 expression remains to be proven. Further, PS proteins are also expressed in neural cells, including glial cells [3,32,92,93,94,95], and previous work indicated a role for these proteins in lipid homeostasis as well as in protein glycosylation in glial cells [32,96,97]. Since it has been clarified that glial lipid dyshomeostasis is associated with their dysfunction and neurodegeneration [98,99,100], further studies utilizing PSs-deficient glial cells could provide us with significant clues to comprehensively understand the effect of PSs dysfunction on neural lipid homeostasis and on neurodegeneration in the brain.

Finally, it is also an important question to ask whether AD-causing PS mutations result in abnormal metabolism of glycoprotein as well as cholesterol. In fact, it has been reported that AD-causing PSs mutants increase cellular cholesterol level [28,33,34]. However, although altered protein glycosylation in the brains of AD patients has been indicated by several studies [72,101,102], it remains unknown whether AD-causing PS mutations, including those of PS2, affect protein glycosylation events. Thus, the effect of the mutants on protein glycosylation and cholesterol metabolism, as well as the causal relationship between the two cellular events, should be evaluated in the future.

## 4. Materials and Methods

### 4.1. Mice

*Presenilin* knockout mice (PS1cKO, PS2KO) [43,44] were housed in humidity- and temperature-controlled rooms and given standard chow and water according to the institute’s guidelines and regulations. Brains were collected from the mice after anesthetization with ketamine (100 mg/kg) + xylazine (10 mg/kg) + acepromazine (3 mg/kg), and transcardial perfusion with phosphate-buffered saline (PBS). The collected brains were immersion-fixed in 4% paraformaldehyde (PFA) in PBS for immunohistochemistry. The fixed brains were cryoprotected in 30% sucrose in PBS for 48 h and snap frozen in isopentane (2-methylbutane) precooled at –80 °C. Sixteen µm sections were then obtained by cryostat, collected in PBS, and stored at 4 °C until analysis. Both male and female mice were used, and the histological analysis experimenter was blind to the genotype of the mice. Animal experiments were approved by the Ethical Committee for Animal Experimentation at the University of Leuven (KU Leuven) (ECD #173/2022) and by the Institutional Animal Care and Use Committees of Brigham and Women’s Hospital and Stanford University.

### 4.2. Cell Culture

Immortalized MEFs [55] and NPC1-KO CHO cells [103] were maintained in culture in Dulbecco’s Modified Eagle’s Medium (DMEM), including GlutaMAX (Thermo Fisher Scientific, Waltham, MA, USA, 10569010) supplemented with 10% heat-inactivated fetal calf serum (FCS) and 1% penicillin/streptomycin solution (50 U/mL penicillin, 50 g/mL streptomycin). Cells were maintained at 37 °C and 5% CO_2_. Treatments with tunicamycin (7 nM) (Sigma, Darmstadt, Germany), 1-Deoxynojirimycin (1 mM) (Cayman chemicals, Ann Arbor, MI, USA), Kifunensin (0.1 µM) (Cayman chemicals, USA), Swainsonine (0.01 µM) (Cayman chemicals, USA), or arimoclomol maleate (300 or 400 µM) (MedChemExpress, Monmouth Junction, NJ, USA) were performed for 6 days, and the medium was changed every other day. The concentration of each inhibitor was determined on the basis of not inhibiting cell proliferation or causing cell death. Cell lines were transfected with the NPC1-His6-EGFP plasmid (RRID: Addgene_53521) using Lipofectamine 2000 Transfection Reagent (Item No. 11668019, Thermo Fisher Scientific), according to the manufacturer’s protocol.

### 4.3. Sample Preparation, SDS-PAGE, and Western Blotting

Cells were collected in cold hypotonic buffer (10 mM Tris-HCl pH 7.4, Complete protease inhibitor (ROCHE, Basel, Switzerland), PhosphoStop (ROCHE)). The nuclear fraction was separated by centrifugation at 1000× *g* for 5 min at 4 °C. Membrane and cytoplasmic fractions were collected from the supernatant by centrifugation at 16,000× *g* for 1 h at 4 °C. Membrane and nuclear fractions were then homogenized by sonication, and protein estimation was performed following the Pierce BCA Protein Assay Kit (Thermo Fisher Scientific, USA) protocol. Samples were mixed with 4× Laemmli buffer (62.5 mM Tris pH 6.8, 2% SDS, 10% glycerol, 5% 2-mercaptoethanol, 0.001% bromophenol blue, 40 mM DTT), subjected to SDS-PAGE, and separated on 4–12% Bis/Tris NuPage gels (Invitrogen, Waltham, MA, USA). After electrophoresis, the proteins were transferred onto a nitrocellulose membrane and detected by Western immunoblotting and ECL imaging. The following antibodies were used for detection: anti-NPC1 (Novus (St. Louis, MO, USA) Cat# NB 400-148, RRID:AB_525790), anti-NPC2 (Sigma-Aldrich Cat# HPA000835, RRID: AB_1079499), anti-Lamp2 (DSHB (Iowa City, IA, USA) Cat# ABL-93, RRID: AB_2134767) (Abcam (Cambridge, UK) Cat# ab37024, RRID: AB_775980), anti-LRP1 (Abcam Cat# ab92544, RRID: AB_2234877), anti-ABCA1 (Novus Cat# NB400-105, RRID: AB_10000630), anti-LDLR (Abcam Cat# ab52818, RRID: AB_881213), anti-SREBP2 (Abcam Cat# ab30682, RRID: AB_779079) (BD Biosciences (Franklin Lakes, NJ, USA) Cat# 557037, RRID: AB_396560), anti-CYP51 (Proteintech (San Diego, CA, USA) Cat# 13431-1-AP, RRID: AB_2088571), anti-N-cadherin (BD Biosciences Cat# 610920, RRID: AB_2077527), anti-nicastrin (BD Biosciences Cat# 612290, RRID: AB_399607), anti-rabbit IgG (Sigma-Aldrich Cat# A9169, RRID: AB_258434), anti-mouse IgG (Sigma-Aldrich Cat# A9044, RRID: AB_258431), anti-rat IgG (Rockland Cat# 612-103-120, RRID: AB_218630).

### 4.4. Immunofluorescence Staining

Cells seeded on coverslips were fixed with 4% PFA for 20 min at RT. The coverslips were then rinsed with PBS and incubated with the blocking solution (10% BSA in PBS), including 100 µg/mL filipin (Sigma-Aldrich Cat# F9765), freshly dissolved in DMSO, for 1 h at RT. Incubation with primary antibodies was carried out overnight at 4 °C. After washing with PBS, the coverslips were incubated with secondary fluorescent-conjugated antibodies for 1 h at RT in the dark. The coverslips were then mounted using an Immunomount solution (Shandon Immu-Mount, Thermo Fisher Scientific, 10662815). All the incubation steps were performed in a humid chamber. In histochemistry, brain sections were transferred onto the coverslips. For β-tubulin III staining, the sections were subjected to heat-induced antigen retrieval using a microwave in sodium citrate buffer (10 mM sodium citrate, 0.05% Tween-20, pH 6.0). Stainings were performed following the same protocol described above except for the filipin concentration, which was increased to 200 µg/mL. The following primary antibodies were used for staining: anti-LAMP2 (DSHB Cat# ABL-93, RRID: AB_2134767), anti-RAB7 (Santa Cruz Biotechnology (Dallas, TX, USA) Cat# sc-376362, RRID:AB_10987863), anti-EEA1 (MBL International (Woburn, MA, USA) Cat# PM062, RRID: AB_10598350), anti-β-tubulin III (Abcam Cat# ab15568, RRID:AB_2210952), anti-NeuN (Sigma-Aldrich Cat# MAB377A5, RRID:AB_2814948), anti-Iba1 (Synaptic Systems (Göttingen, Germany) Cat# 234 003, RRID: AB_10641962), anti-GFAP (Cell Signaling Technology (Danvers, MA, USA) Cat# 12389, RRID: AB_2631098). The following fluorescent-conjugated secondary antibodies were used: Donkey anti-rabbit 546 (Thermo Fisher Scientific Cat# A10040, RRID: AB_2534016), donkey anti-mouse 488 (Thermo Fisher Scientific Cat# A-21202, RRID: AB_141607), goat anti-rabbit 546 (Thermo Fisher Scientific Cat# A-11010, RRID: AB_2534077), goat anti-rabbit 647 (Thermo Fisher Scientific Cat# A-21244, RRID: AB_2535812), goat anti-mouse 488 (Thermo Fisher Scientific Cat# A-11001, RRID: AB_2534069), goat anti-mouse 546 (Thermo Fisher Scientific Cat# A-11003, RRID: AB_2534071), goat anti-rat 546 (Thermo Fisher Scientific Cat# A-11081, RRID: AB_2534125).

### 4.5. Imaging and Data Analysis

Images were acquired using the Zeiss Axio Vert series Axiovert 200 inverted microscope (RRID: SCR_020915) and ZEN microscopy software, ZEN 3.0 (Zeiss, Oberkochen, Germany) or the VisiScope CSU-W1 spinning disk confocal microscope and VisiView software, version 4 (Visitron System GmbH, Puchheim, Germany). Filipin fluorescence was detected using UV excitation at 350 nm. In Figure 1, detection of TUJ1 and NeuN was obtained using the secondary fluorescent antibodies goat anti-rabbit 546 and goat anti-mouse 488, respectively. An analysis of filipin intensity was performed in TUJ1/NeuN double-positive neurons. In particular, a region of interest (ROI) was manually drawm to select the TUJ1 signal surrounding the NeuN-positive nuclei. The same ROI was then used to measure the intensity in the Filipin channel. Detection of Lamp2 was obtained using the secondary antibody goat anti-rat 546, while for RAB7 and EEA1, goat anti-mouse 488 and goat anti-rabbit 647 were used, respectively. The laser’s power and exposure time settings were maintained constant through the acquisition of experimental group samples and respective controls. For each experimental group, at least six images per coverslip were randomly taken and used for quantification. For all fluorescence microscopy images, acquisition and quantification were performed using 8 bit TIFF images in grey scale. For the presentation of composite pictures, the green and red channels were chosen as pseudo-colors to visualize the staining of interest. Image processing was performed using the free software ImageJ (Version: 2.9.0/1.53t, RRID: SCR_003070).

### 4.6. Sterol Mass Spectrometry Analysis

MEFs cultivated until confluent were rinsed using cold PBS, harvested with Hanks’ Balanced Salt Solution (HBSS) (Thermo Fisher Scientific, 14170112), and centrifuged at 1000× *g* for 3 min at 4 °C. The cell pellet was snap frozen and kept at −80 °C until analysis. The cell pellet was applied for mass spectrometry (MS) for quantitative analysis of sterols as described previously [32].

### 4.7. N-Glycan Structural Analysis by Mass Spectrometry

Whole cell lysates were prepared from MEFs collected by lysing in Tris-buffered saline (TBS) (50 mM tris(hydroxymethyl)aminomethane, 150 mM NaCl, pH 7.4), containing 1% Triton X-100, 1 mM EDTA, and a protease inhibitor cocktail (cOmplete^TM^, Roche, Basel, Switzerland) as described previously [104,105]. N-glycans were released from the whole-cell lysate protein (25 μg) with 2 units of PNGase F (Roche) after reductive alkylation and trypsin digestion. The released N-glycans were captured and labeled with N^α^-((aminooxy)acetyl)tryptophanylarginine methyl ester (aoWR) by BlotGlyco beads (Sumitomo Bakelite, Tokyo, Japan) as described previously [106,107]. After removal of excess reagents by MassPrep HILIC μElution plate, matrix-assisted laser desorption/ionization-time of flight mass spectrometry (MALDI-TOF MS) analyses of aoWR-labeled glycans were performed on Autoflex Speed (Bruker Daltonics, Billerica, MA, USA) operated in positive-ion reflector mode. For MS acquisition, aoWR-labeled glycans in acetonitrile were mixed 1:1 with dihydrobenzoic acid (10 mg/mL in 50% acetonitrile) and spotted on the target plate.

### 4.8. Deglycosylation of Membrane Proteins

Enzymatic deglycosylation of membrane proteins was performed using PNGase F (New England Biolabs, Ipswich, MA, USA, P0704) according to the manufacturer’s protocol. Deglycosylated samples were applied for SDS-PAGE and Western blotting analysis.

### 4.9. 3′-mRNA Sequencing

RNA extraction from two independent preparations of MEFs was performed using Trizol (Invitrogen) following the manufacturer’s instructions. 3′-mRNA sequencing was performed in the Next Generation Sequencing (NGS) Core Facility of the Universitätsklinikum Bonn. Libraries were prepared using the QuantSeq 3′-mRNA-Seq Fw. Library Prep Kit (Lexogen, Greenland, NH, USA). Quality control was performed using fastqc v0.11.8 (Babraham Bioinformatics Institute, Cambridge, UK), followed by trimming using bbduk. Alignment was performed with STAR aligner 2.6.0a against the GRCm38 mouse (Ensembl release 102) genome. The R-package DESeq2 was used for statistical analysis with settings as recommended by the provider (normalization of raw counts, dispersion estimation, and negative binomial Wald test with Benjamini-Hochberg multiple test correction). Adjusted *p* values < 0.05 were defined as significant. The NGS data was deposited in GEO (GSE223572).

### 4.10. Statistical Analysis

Ordinary one-way ANOVA followed by Dunnett’s correction of the two-tailed Student’s *t* test were used for statistical analysis using GraphPad Prism (RRID:SCR_002798). The data are represented as the mean ± standard error of the mean (SEM).

## Figures and Tables

**Figure 1 ijms-25-05417-f001:**
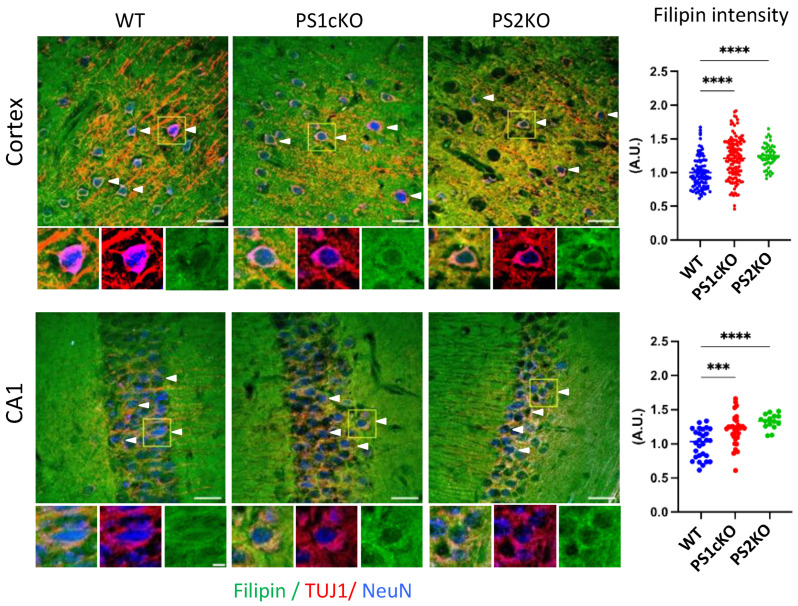
Neuronal cholesterol accumulation in PS knockout mouse brains. Representative co-immunostaining of filipin (green) with β3-tubulin (TUJ1, red) and NeuN (blue) in the cortex and CA1 region from wild-type (WT), PS1-conditional knockout (PS1cKO), and PS2KO mice. Three (PS2KO) or six brains (WT, PS1cKO) from each genotype were analyzed. At least four images per brain region were taken from each sample. An analysis of filipin intensity was performed in TUJ1/NeuN double-positive neurons (arrow heads). The original 8-bit pictures were pseudo colored to improve visualization. Scale bar: 50 µm. Graphs show the filipin intensity quantified in TUJ1/NeuN double-positive cells; one-way ANOVA with Dunnett’s correction; *** *p* < 0.001; **** *p* < 0.0001.

**Figure 2 ijms-25-05417-f002:**
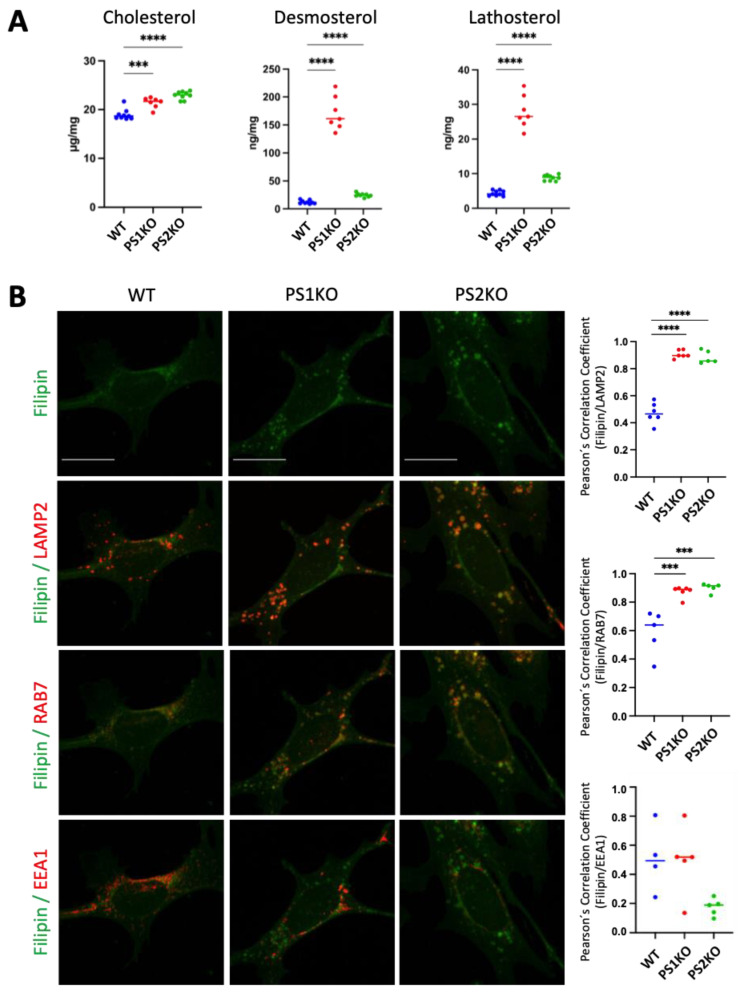
Cholesterol accumulation in PS1 and PS2 KO mouse embryonic fibroblasts. (**A**) Cholesterol and its precursors, desmosterol and lathosterol, are measured by mass spectrometry in mouse embryonic fibroblasts (MEFs) or wild-type (WT), PS1 (PS1KO), and PS2 KO (PS2KO). Brown-Forsythe and Welch ANOVA multiple comparison test, Dunnett correction; ***, *p* < 0.0005; ****, *p* < 0.0001. (**B**) Representative co-immunostaining of filipin (green) with LAMP2, RAB7, or EEA1 (red). The original 8-bit pictures were pseudo colored in green and red to better visualize colocalization. Four to six pictures per sample were taken, and the mean value of the quantified cells per picture is shown in the graph as one data point. Scale bar: 20 μm. Graphs show the Pearson’s correlation coefficient obtained from three independent preparations (N = 3); Ordinary one-way ANOVA with Dunnett’s correction; ***, *p* < 0.0005; ****, *p* < 0.0001.

**Figure 3 ijms-25-05417-f003:**
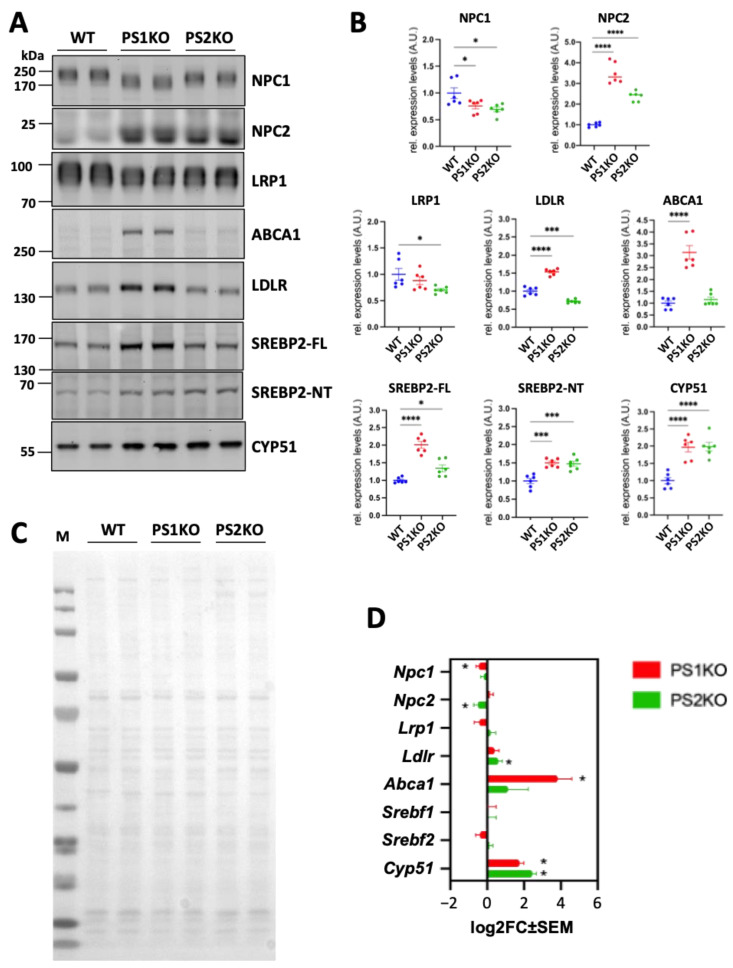
Abnormal expression of proteins involved in cholesterol metabolism in PS1 and PS2 KO mouse embryonic fibroblasts. (**A**) Representative western blotting of proteins involved in cellular cholesterol metabolism; and (**B**) relative quantification. Signal intensities were normalized to the signals of Ponceau staining (**C**). Values were obtained from three independent experiments with two biological replicates (N = 3). FL, full length; NT, N-terminal; M, protein molecular weight marker; A.U., arbitrary units; ordinary one-way ANOVA with Dunnett’s correction; *, *p* < 0.05; ***, *p* < 0.0005; ****, *p* < 0.0001. (**D**) Graph of differentially expressed genes in PS KO MEFs (average of 5 samples) compared to WT (average of seven samples) in mRNA sequencing analysis. The log2FC (fold change) ± SEM of the genes in PS1KO or PS2KO MEFs are shown. Adjusted *p* value < 0.05, binominal Wald test, followed by Benjamini-Hochberg correction.

**Figure 4 ijms-25-05417-f004:**
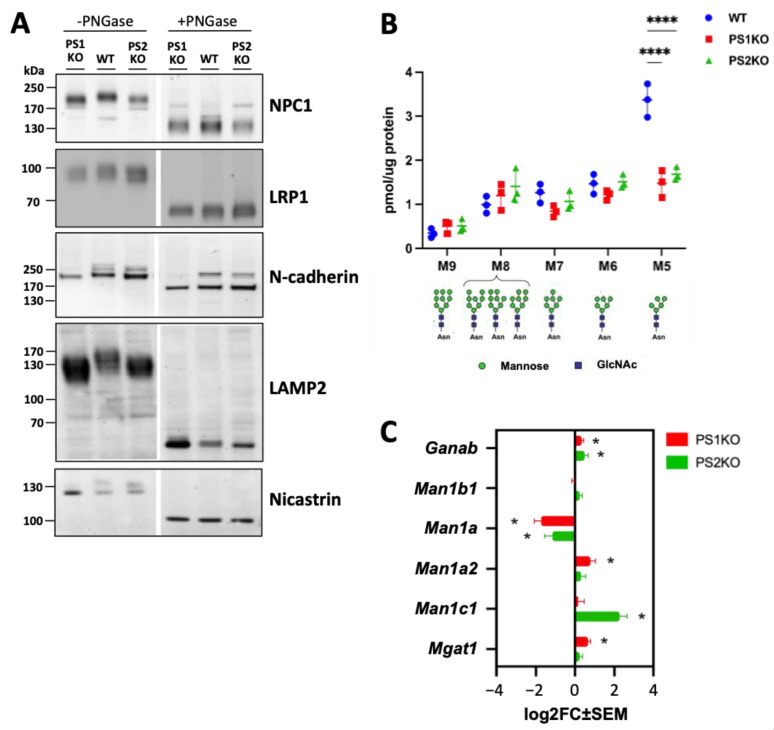
Membrane protein glycosylation is impaired in PS1 and PS2 KO mouse embryonic fibroblasts. (**A**) Membrane fraction samples from MEFs WT and PS KO were exposed to PNGase or water (controls, -PNGase) and applied for western blotting to evaluate the migration of the bands after complete cleavage of the glycans. Corresponding parts are cropped from the original blots and combined in each image (Appendix A). (**B**) Graph showing the amount of specific N-glycan structure measured by mass spectrometry. The results are from three independent sample preparations (N = 3). Ordinary two-way ANOVA with Dunnett’s correction. ****, *p*< 0.0001. (**C**) A graph shows differentially expressed genes in PS KO MEFs (average of 5 samples) compared to WT (average of seven samples) in mRNA sequencing analysis. The log2FC (fold change) ± SEM of the genes in PS1KO or PS2KO MEFs are shown. Adjusted *p* value < 0.05, binominal Wald test, followed by Benjamini-Hochberg correction. *, *p*< 0.05.

**Figure 5 ijms-25-05417-f005:**
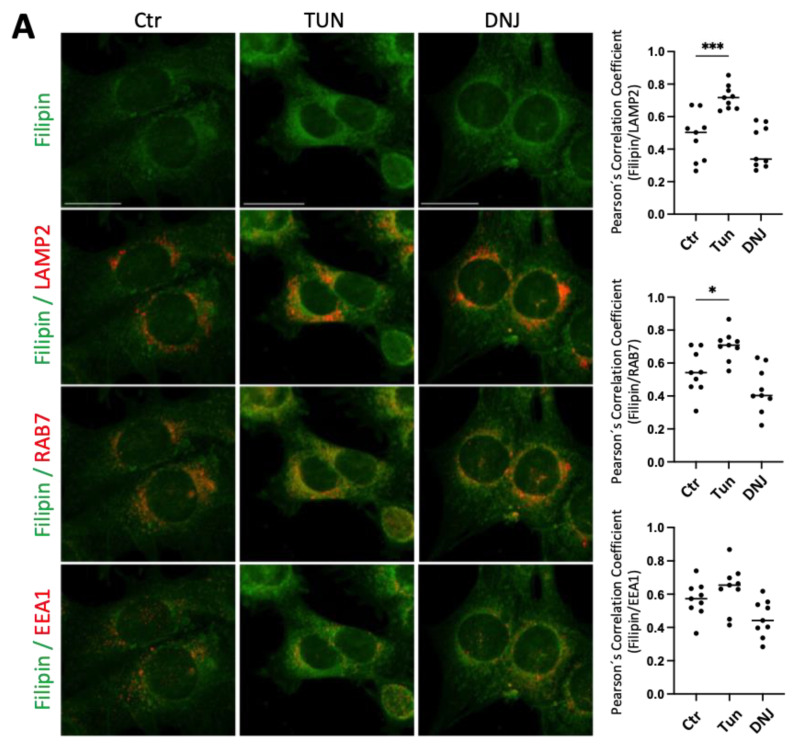
Cholesterol subcellular distribution upon treatments with glycosylation inhibitors in wild-type mouse embryonic fibroblasts. Representative co-immunostaining of filipin (green) with LAMP2, RAB7, or EEA1 (red) in wild-type (WT) mouse embryonic fibroblasts (MEFs) untreated (Ctr) or treated with Tunicamycin (TUN) and Deoxynojirimycin (DNJ) (**A**) or Kifunensin (KIF) and Swainsonine (SW) (**B**). Scale bar: 20 μm. Nine pictures per sample were taken, and the mean value of the quantified cells per picture is shown in the graph as one data point. The original 8-bit pictures were pseudo colored in green and red to better visualize colocalization. Graphs show the Pearson’s correlation coefficient obtained from three independent preparations (N = 3); ordinary one-way ANOVA with Dunnett’s correction; *, *p* < 0.05; ***, *p* < 0.0005.

**Figure 6 ijms-25-05417-f006:**
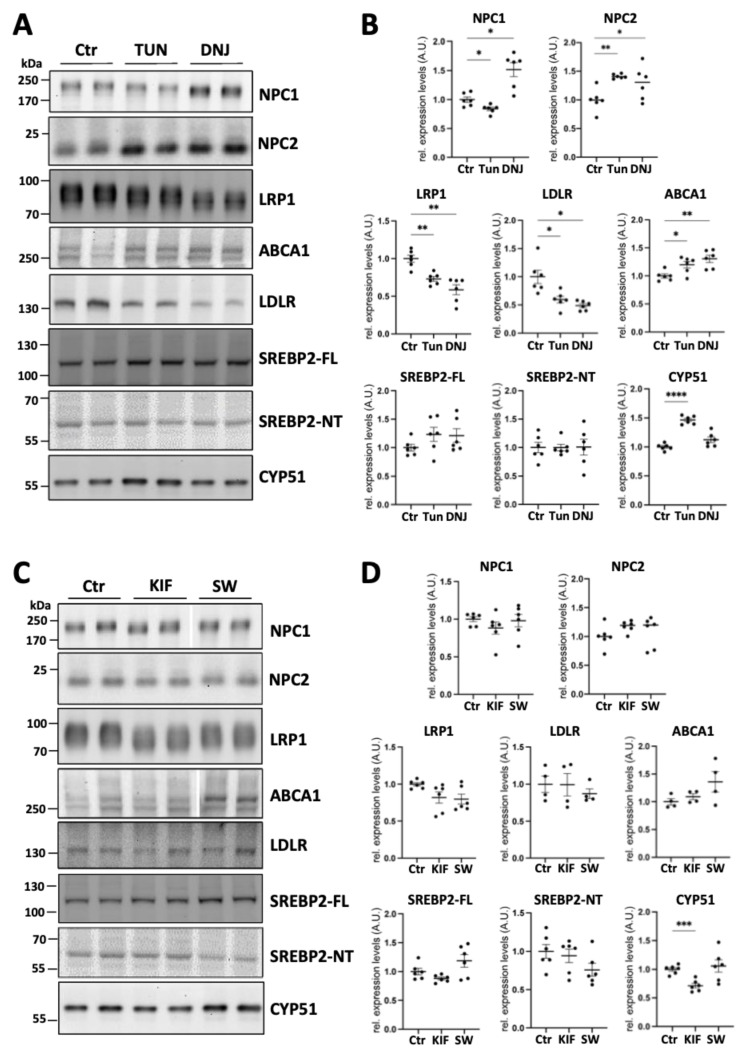
Impairment of protein glycosylation by inhibitors affects the expression of proteins involved in cellular cholesterol metabolism in wild-type mouse embryonic fibroblasts. (**A**) Representative western blotting of the effect of Tunicamycin (TUN) and Deoxynojirimycin (DNJ) treatments on proteins related to cholesterol metabolism in wild-type (WT) mouse embryonic fibroblasts (MEFs) and (**B**) relative quantification. (**C**) Representative western blotting showing the effect of Kifunensin (KIF) and Swainsonine (SW) treatments on proteins related to cholesterol metabolism, and (**D**) relative quantification. Corresponding parts are cropped from the original blots and combined in images of NPC1 and ABCA1 (Appendix A). Signal intensities were normalized to Ponceau. Values were obtained from three independent experiments with two biological replicates (N = 3). CTR, control; AU, arbitrary units; Ordinary one-way ANOVA with Dunnett’s correction; *, *p* < 0.05; **, *p* < 0.005; ***, *p* < 0.0005; ****, *p* < 0.0001.

**Figure 7 ijms-25-05417-f007:**
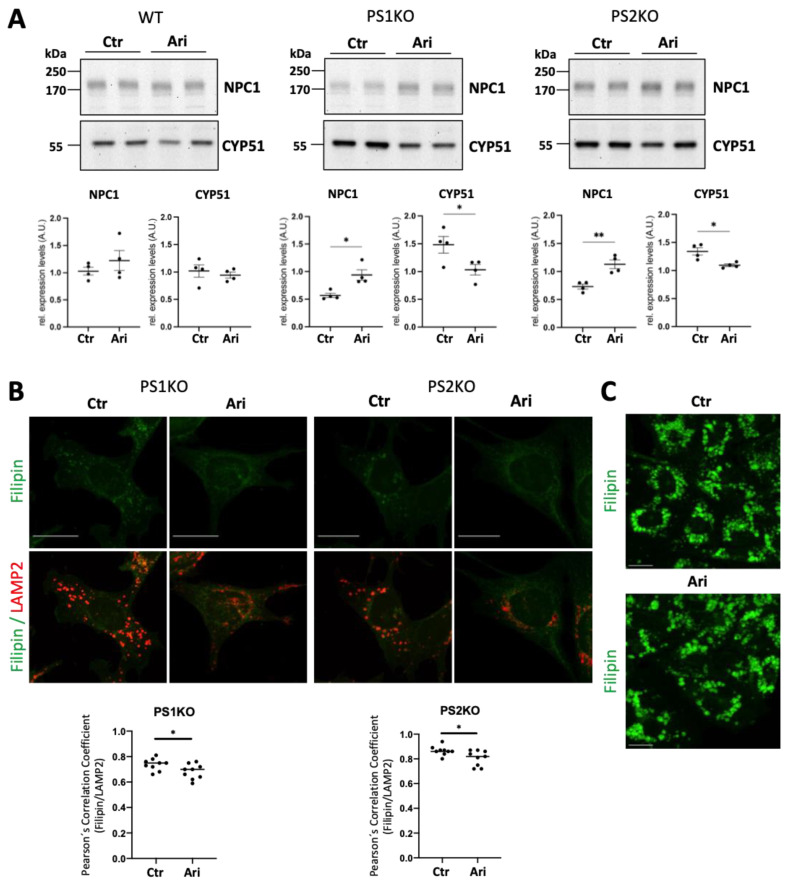
Chaperon inducer treatment increases NPC1 expression and reduces lysosomal cholesterol accumulation in PS1 and PS2 KO mouse embryonic fibroblasts. (**A**) Representative western blotting showing the effect of Arimoclomol maleate (Ari) treatments on protein expression in PS1 (PS1KO) and PS2 KO (PS2KO) mouse embryonic fibroblasts (MEFs). The graphs show the results of four independent experiments (N = 4). Unpaired two-tailed Student’s *t* test. *, *p* < 0.05; **, *p* < 0.01. (**B**) Representative co-immunostaining of filipin (green) with LAMP2 (red) in PS1KO and PS2KO MEFs untreated (Ctr) or treated with Ari. The original 8-bit pictures were pseudo colored in green and red to better visualize colocalization. Nine pictures per sample were taken, and the mean value of the quantified cells per picture is shown in the graph as one data point. Scale bar: 20 μm. (**C**) Representative filipin staining (green) of NPC1-KO Chinese hamster ovary (CHO) cells (CHO) untreated (Ctr) or treated with Ari. Scale bar: 20 μm.

**Figure 8 ijms-25-05417-f008:**
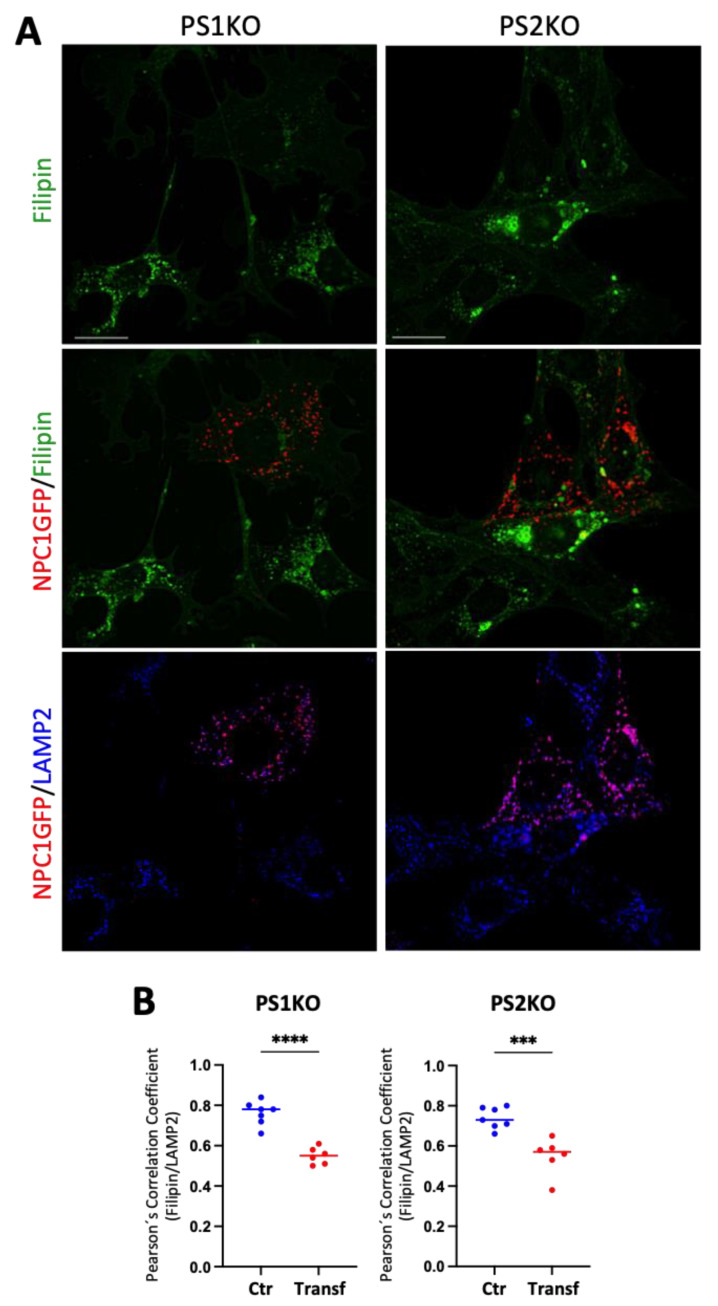
NPC1 overexpression rescues the lysosomal cholesterol accumulation in PS1 and PS2 KO mouse embryonic fibroblasts. (**A**) Representative immunostaining of filipin (green) with the NPC1-His6-EGFP construct (red) and LAMP2 (blue) in PS1 (PS1KO) and PS2KO (PS2KO) mouse embryonic fibroblasts (MEFs). Scale bar: 20 μm. (**B**) Colocalization analysis of the filipin signal and LAMP2 in transfected and non-transfected (Ctr) cells. Two coverslips per condition were analyzed from three independent transfections (N = 3). Data points on graphs represent individual quantified cells. The original 8-bit pictures were pseudo colored to better visualize the visualization. Unpaired two-tailed Student’s *t* test. ***, *p* < 0.0005; ****, *p* < 0.0001.

## Data Availability

The raw data used in this study are available upon reasonable request.

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
