# Peer review of "Presenilin Deficiency Results in Cellular Cholesterol Accumulation by Impairment of Protein Glycosylation and NPC1 Function"

_ijms, 2024, doi:10.3390/ijms25105417_

Round 1

Reviewer 1 Report

Comments and Suggestions for Authors

The manuscript by Fabiano et al. reports that presenilin deficiency leads to cellular cholesterol accumulation through impairment of protein glycosylation and NPC1 function. Thus, the manuscript represents an elegant, well-designed study in which the conclusions are well supported by the experimental evidence. I have a number of concerns, all related to the presentation: 

- Figure 1 in the version I have is of low quality. I also think it would be nice to enlarge the panels representing the graphs. 

- Similar to Figs. 2&5: the graphs could be enlarged. 

Author Response

We highly appreciate your kind and helpful comments. We have addressed your concerns.

Point 1: Figure 1 in the version I have is of low quality. I also think it would be nice to enlarge the panels representing the graphs.

-Response: We apologize for our uploading the low-resolution images. We have replaced the presented images with high-resolution images not only in figure 1 but also in other figures.

Point 2: Similar to Figs. 2&5: the graphs could be enlarged.

-Response: According to the comment, we have enlarged the graphs not only in the indicated figures but also in other figures.

Reviewer 2 Report

Comments and Suggestions for Authors

The present study explores the roles of both presenilin 1 and 2 in cellular cholesterol metabolism. The authors used specific knock-out animal models and cellular and molecular biology techniques trying to associate their results with the expression levels of NPC1 protein. It is very interesting that the loss of presenilin homology lead in protein glycosylation impairment and at the same time cholesterol accumulation. The manuscript is well-written. Some minor comments could be taken into account for improvement that I would like to be addressed.

The authors could increase the quality of Figures.

Limitations of the present process and some future directions could be also discussed.

Author Response

We highly appreciate your kind and helpful comments. We have addressed your concerns as follows.

Point 1: The authors could increase the quality of Figures.

-Response: We apologize for our uploading the low-resolution images. We have replaced the presented figures with high-resolution ones.

Point 2: Limitations of the present process and some future directions could be also discussed.

-Response: According to the comment, we have added a paragraph regarding the indicated points in the Discussion section with updating the reference list.

“As the present study only applied PS-KO MEFs as cellular model to characterize molecular mechanisms underlying the intracellular cholesterol accumulation caused by PS deficiency, it would be important to validate the present findings in different cell types of the brain, including neuronal and glial cells. Although our data revealed cholesterol accumulation in cortical and CA1 hippocampal neurons of PS1cKO and PS2KO mice, the involvement of protein N-glycosylation and NPC1 expression remains to be proven. Further, PS proteins are also expressed in neural cells including glial cells [3,32,92,93,94,95], and previous work indicated a role of these proteins in lipid homeostasis as well as in protein glycolsylation in glial cells [32,96,97]. Since it has been clarified that glial lipid dyshomeostasis is associated with their dysfunction and neurodegeneration [98,99,100], further studies utilizing PSs-deficient glial cells could provide us significant clues to comprehensively understand the effect of PSs dysfunction on neural lipid homeostasis and on neurodegeneration in brain.”

92.           Farfara, D.; Trudler, D.; Segev-Amzaleg, N.; Galron, R.; Stein, R.; Frenkel, D. γ-Secretase Component Presenilin Is Important for Microglia β-Amyloid Clearance. Ann Neurol 2011, 69, 170–180.

93.           Glebov, K.; Wunderlich, P.; Karaca, I.; Walter, J. Functional Involvement of γ-Secretase in Signaling of the Triggering Receptor Expressed on Myeloid Cells-2 (TREM2). J Neuroinflammation 2016, 13, 17.

94.           Walter, J.; Kemmerling, N.; Wunderlich, P.; Glebov, K. γ-Secretase in Microglia - Implications for Neurodegeneration and Neuroinflammation. J Neurochem 2017, 143, 445–454.

95.           Hou, P.; Zielonka, M.; Serneels, L.; Martinez-Muriana, A.; Fattorelli, N.; Wolfs, L.; Poovathingal, S.; T’Syen, D.; Balusu, S.; Theys, T.; et al. The γ-Secretase Substrate Proteome and Its Role in Cell Signaling Regulation. Mol Cell 2023, 83, 4106-4122.e10.

96.           Islam, S.; Sun, Y.; Gao, Y.; Nakamura, T.; Noorani, A.A.; Li, T.; Wong, P.C.; Kimura, N.; Matsubara, E.; Kasuga, K.; et al. Presenilin Is Essential for ApoE Secretion, a Novel Role of Presenilin Involved in Alzheimer’s Disease Pathogenesis. J Neurosci 2022, 42, 1574–1586.

97.           Chen, F.; Tandon, A.; Sanjo, N.; Gu, Y.-J.; Hasegawa, H.; Arawaka, S.; Lee, F.J.S.; Ruan, X.; Mastrangelo, P.; Erdebil, S.; et al. Presenilin 1 and Presenilin 2 Have Differential Effects on the Stability and Maturation of Nicastrin in Mammalian Brain. J Biol Chem 2003, 278, 19974–19979.

98.           Guttenplan, K.A.; Weigel, M.K.; Prakash, P.; Wijewardhane, P.R.; Hasel, P.; Rufen-Blanchette, U.; Münch, A.E.; Blum, J.A.; Fine, J.; Neal, M.C.; et al. Neurotoxic Reactive Astrocytes Induce Cell Death via Saturated Lipids. Nature 2021, 599, 102–107.

99.           Marschallinger, J.; Iram, T.; Zardeneta, M.; Lee, S.E.; Lehallier, B.; Haney, M.S.; Pluvinage, J.V.; Mathur, V.; Hahn, O.; Morgens, D.W.; et al. Lipid-Droplet-Accumulating Microglia Represent a Dysfunctional and Proinflammatory State in the Aging Brain. Nat Neurosci 2020, 23, 194–208.

100.         Haney, M.S.; Pálovics, R.; Munson, C.N.; Long, C.; Johansson, P.K.; Yip, O.; Dong, W.; Rawat, E.; West, E.; Schlachetzki, J.C.M.; et al. APOE4/4 Is Linked to Damaging Lipid Droplets in Alzheimer’s Disease Microglia. Nature 2024, 628, 154–161.

Reviewer 3 Report

Comments and Suggestions for Authors

The paper of Fabiano et al, entitled "Presenilin deficiency results in cellular cholesterol accumulation by impairment of protein glycosylation and NPC1 function" addresses a very serious public health problem, i.e. Alzheimer disease. The authors studied PS1 and PS2 that play a critical role in Amyloid beta generation in correlation with cholesterol metabolism in cerebral neurons and embryonic fibroblasts from PS1-KO and PS2-KO mice. The authors proved that cholesterol accumulation was correlated with a lower expression of NPC1 and an abnormal N-glycosylation of NPC1, LRP1, N-cadherin, LAMP2 and nicastrin. In fact they proved that impairment of N-glycosylation induced cholesterol  accumulation in brain.

The authors used different biochemical, cell biology, mass spectrometry and mRNA seq analyses.

Also, the references are chosen appropriately.

Author Response

Thank you very much for your kind reviewing and comments.

Since we have realized that quality and size of images and graphs can be increased and enlarged, respectively, we have exchanged the presented figures with improved ones.

Reviewer 4 Report

Comments and Suggestions for Authors

This is a well-written manuscript "entitled Presenilin deficiency results in cellular cholesterol accumulation by impairment of protein glycosylation and NPC1 function". 

The work is well illustrated. The conclusions are supported by the results. The work itself is of great importance for understanding the mechanisms of impaired lipid metabolism, and its relationship to some diseases, especially Alzheimer's disease.

I believe that the manuscript can be published in its current form.

Author Response

(The authors gave the same response as above.)
